# A Comprehensive Review of the Current Status of the Cellular Neurobiology of Psychedelics

**DOI:** 10.3390/biology12111380

**Published:** 2023-10-28

**Authors:** Blerida Banushi, Vince Polito

**Affiliations:** 1Genetics and Genomic Medicine Department, Great Ormond Street Institute of Child Health, University College London, London WC1N 1EH, UK; 2School of Psychological Sciences, Macquarie University, Sydney, NSW 2109, Australia; vince.polito@mq.edu.au

**Keywords:** psychedelics, 5-HT2A, BDNF, TrkB, serotonergic, psilocybin, LSD, psychedelic therapy, hallucinogen, neuroplasticity

## Abstract

**Simple Summary:**

Understanding the cellular neurobiology of psychedelics is crucial for unlocking their therapeutic potential and expanding our understanding of consciousness. This review provides a comprehensive overview of the current state of the cellular neurobiology of psychedelics, shedding light on the intricate mechanisms through which these compounds exert their profound effects. Given the significant global burden of mental illness and the limited efficacy of existing therapies, the renewed interest in these substances, as well as the discovery of new compounds, may represent a transformative development in the field of biomedical sciences and mental health therapies.

**Abstract:**

Psychedelic substances have gained significant attention in recent years for their potential therapeutic effects on various psychiatric disorders. This review delves into the intricate cellular neurobiology of psychedelics, emphasizing their potential therapeutic applications in addressing the global burden of mental illness. It focuses on contemporary research into the pharmacological and molecular mechanisms underlying these substances, particularly the role of 5-HT2A receptor signaling and the promotion of plasticity through the TrkB-BDNF pathway. The review also discusses how psychedelics affect various receptors and pathways and explores their potential as anti-inflammatory agents. Overall, this research represents a significant development in biomedical sciences with the potential to transform mental health treatments.

## 1. Introduction

Coined by Humphry Osmond in 1956, the term “psychedelic” originates from the Greek words meaning “mind manifesting” [1]. This term is used to describe the subjective effects of these substances, highlighting their ability to induce profound experiences and alter perception.

Psychedelics constitute a class of drugs obtained from specific plants, animals, and fungi, and they can be categorized into three primary classes based on their chemical structure: tryptamines, ergolines, and phenethylamines [2,3]. Tryptamines, such as psilocybin, N, N-dimethyltryptamine (DMT), and 5-methoxy-N,N-dimethyltryptamine (5-MeO-DMT) are characterized by an indole (aromatic group separated from a basic amine by a two-carbon linker) and share structural similarities with the neurotransmitter serotonin. Ergolines, such as lysergic acid diethylamide (LSD), are characterized by the presence of a tetracyclic ergoline ring, and were originally derived from the ergot fungus [4]. Phenethylamines, such as 2C-B, mescaline, amphetamine analogues—e.g., 2,5-Dimethoxy-4-iodoamphetamine (DOI) and 2,5-Dimethoxy-4-methylamphetamine (DOM), and derivatives such as 4-Iodo-2,5-dimethoxy-N-(2-methoxybenzyl) phenethylamine (25I-NBOMe)—are characterized by a benzene ring with an amino group attached through a two-carbon chain [3]. In addition to classical psychedelics, there are atypical compounds like 3,4-Methylenedioxymethamphetamine (MDMA), muscimol, scopolamine, salvinorin A, ibogaine, nitrous oxide, phencyclidine (PCP), and ketamine that produce similar psychological effects but work through different mechanisms. These compounds are sometimes considered psychedelics under a broader definition [4].

Psychedelics have been used by humans for centuries. Historical records indicate that these substances have been consumed in ancient cultural rituals with the purpose of healing, attaining altered states of consciousness, and gaining spiritual insights, tracing back to prehistory [5,6]. The synthesis of mescaline in the early 1900s and the groundbreaking discovery of the classical hallucinogen LSD by Albert Hofmann in 1938 marked the beginning of Western psychedelic science [7,8,9]. During the 1950s and 1960s, these substances gained popularity in therapeutic and psychiatric settings due to their ability to facilitate psychotherapy [7]. However, in 1970, the U.S. Drug Enforcement Agency classified psychedelics as Schedule I drugs, which had a profound impact on research in the field [10]. Prior to their classification, over 1000 clinical studies were published, documenting promising therapeutic effects of psychedelics in more than 40,000 subjects. [7,11,12]. These studies indicated therapeutic benefits in various conditions such as anxiety and obsessive-compulsive disorders (OCD), depression, alcohol addiction, and sexual dysfunction, as well as pain and anxiety relief in patients with terminal cancer [10,13,14,15,16,17,18,19]. This regulatory decision effectively curtailed psychedelic research for approximately 30 years [20].

The advent of advanced neuroimaging techniques in the 1990s, including positron emission tomography (PET) and functional magnetic resonance imaging (fMRI), played a pivotal role in enhancing our comprehension of molecular and physiological mechanisms within the central nervous system [21]. These breakthroughs also sparked a renewed interest in exploring the effects of psychedelic substances [22]. In the last three decades, there has been a resurgence in the exploration of psychedelics, reigniting research interest in their therapeutic potential. This renewed focus has led to anticipated FDA approvals for the use of psychedelics in treating various conditions, marking a period of exponential scientific growth in this field [23,24,25].

In recent years a growing number of clinical trials and studies have been conducted or are currently underway, investigating the therapeutic potential of psychedelics such as LSD and psilocybin for various mental health conditions including depression, anxiety, cancer-related anxiety disorders, addiction, post-traumatic stress disorder (PTSD), obsessive-compulsive disorder, terminal illness, stroke, traumatic brain injury (TBI), neurodegenerative disorders, and chronic pain [22,26,27,28,29,30,31,32,33,34,35,36,37,38,39,40,41]. In response to this growing interest, the U.S. Food and Drug Administration recently released a new draft guidance aimed at emphasizing essential considerations for researchers exploring the use of psychedelic drugs as potential treatments for medical conditions [42].

Randomized Phase II trials have so far demonstrated significant reductions in symptoms and long-lasting benefits following psilocybin-assisted psychotherapy for major depressive disorder, anxiety, and treatment-resistant depression [43,44,45,46]. Remarkably, even one or two doses of psilocybin have led to rapid and sustained improvements in mood and perspective, with symptomatic relief lasting for at least 3–12 months [28,45,46,47]. Furthermore, enduring positive effects and improved well-being following psilocybin administration have also been observed in healthy individuals. Recent Phase 3 trials investigating MDMA for PTSD have shown promising results, and hint at a potential paradigm shift in psychiatry towards utilizing substances with acute psychoactive effects to generate long-term benefits for psychiatric patients [48].

Understanding the cellular neurobiology underlying these mind-altering substances is crucial for unraveling their therapeutic potential and expanding our knowledge of consciousness. This review aims to provide a comprehensive overview of the current status of the cellular neurobiology of psychedelics, shedding light on the intricate mechanisms through which these compounds exert their profound effects. Given the significant societal and economic burden associated with mental illness globally [49], the limited efficacy of many existing therapies, and the transdiagnostic potential of psychedelics, the renewed interest in these natural substances, as well as the discovery of new semisynthetic and synthetic compounds, may represent one of the most transformative advancements in the biomedical sciences.

## 2. Psychedelics Exert Their Effects on the Brain at Multiple Levels, Engaging in Intricate and Multifaceted Mechanisms

The impact of psychedelics on the brain can be examined from multiple perspectives, including molecular/cellular, circuit/network, and overall brain levels, all of which are inherently interconnected (Figure 1). At the molecular/cellular level, psychedelics stimulate the serotonin 2A receptor (5-HT2R) along with other serotonin sub-receptors, tropomyosin receptor kinase B (TrkB), and dopamine receptors [50,51]. Classic psychedelics have been shown to elevate levels of glutamate and oxytocin [22,52,53,54,55], promote the production of brain-derived neurotrophic factor (BDNF) [51,56,57], stimulate neurogenesis [22,52], and exhibit anti-inflammatory properties [58]. On a cellular level, psychedelics also induce an increase in the expression of various genes that encode for the synthesis of a range of proteins that facilitate neuroplasticity and learning, even following a single dose [59,60,61].

At the circuit and network levels (Figure 1), both ketamine and serotonergic psychedelics have been observed to enhance synaptic growth and increase the complexity of dendrites, consequently leading to a greater number of synapses [66]. This, in turn, results in an increased connectivity among neurons [67,68,69,70,71,72,73]. 

The structural and neuroplastic changes induced by psychedelics originate from a series of processes discussed earlier at the molecular and cellular levels. This review will shed light on some of these mechanisms. This cascade of network changes has led to psychedelics being characterized as “psychoplastogens”, substances with the capability to facilitate rapid neural plasticity both in terms of structure and function [67,74].

At the level of the brain (Figure 1), several distinct and frequently complementary neuroscientific explanations have been put forth to elucidate the effects of psychedelics and the mechanisms underlying the psychedelic experience [75,76]. The three most prominent theories encompass the cortico-striato-thalamo-cortical (CSTC) model [63], the relaxed beliefs under psychedelics (REBUS) model [64], and the claustro-cortical circuit (CCC) model [65].

The CSTC model proposes that psychedelics, through the stimulation of 5-HT2ARs, disrupt information processing in the brain by altering the thalamo-cortical gating of external and internal information to the cortex [63]. This disruption leads to an increased flow of information or feedforward processing, resulting in various effects, including impaired sensorimotor gating, alterations in cognitive functioning, and changes in sensory and somatomotor cortical regions. This model is substantiated by both behavioral observations of impaired sensorimotor gating in humans following the administration of psilocybin [77,78], LSD [79] and ayahuasca [80], and neuroimaging findings demonstrating increased thalamic functional connectivity and synchronisation of cortical sensory regions in response to LSD [81,82].

The REBUS model suggests that psychedelics reduce the precision of high-level priors (or expectations and beliefs about the world) while simultaneously increasing the flow of bottom-up sensory information [64]. This model integrates the entropic brain hypothesis [83] and the free-energy principle, explaining how psychedelics influence brain function by altering the relative importance of prior beliefs and sensory information. This shift in signal weighting makes recurrent message transfer within the brain more responsive to modulation by incoming sensory signals, ultimately leading to increased complexity and entropy in neuronal dynamics. Preliminary empirical evidence supports this model, demonstrating that psychedelics like LSD, psilocybin, DMT, ketamine, and ayahuasca enhance signal diversity and measures of entropy in brain activity [84,85,86,87,88]. While the REBUS model has influenced the field of psychedelic research as a potential overarching framework, it has faced criticism both conceptually (such as regarding entropy definition, low- and high-level brain region demarcation) and methodologically (small sample sizes of the studies and analytical choices), necessitating further rigorous research for confirmation [75,76]. 

The CCC model proposes that psychedelics interfere with coordination between cortical regions and the claustrum by directly activating 5-HT2ARs, which are abundant in the claustrum [65,89,90,91]. This coordination plays a crucial role in cognitive control, which is diminished by the effects of psychedelics [65,91]. This model has substantial support from neuroimaging studies that have demonstrated that psilocybin significantly disrupts networks associated with cognitive control and the proper functioning of the claustrum [68]. While the CCC model holds promise in explaining the widespread effects of psychedelics on various brain networks, it currently lacks specificity in detailing how these changes in the claustrum affect specific canonical circuits [75]. Advanced imaging techniques, such as higher field fMRI, will be needed to better understand the flow of information between the claustrum and other brain regions. Additionally, although reduced cognitive control may contribute to some psychedelic effects, it appears insufficient to explain all of the varied acute subjective experiences associated with these substances [75].

The models and levels of analysis presented in Figure 1 include extra-pharmacological factors, such as social, contextual, and cultural elements, commonly referred to as “set” (comprising individual beliefs, expectations, and mindset) and “setting” (involving the environment and socio-historical context). Of particular note, music has consistently played a pivotal role in guiding and enhancing the therapeutic experience in the history of psychedelic research, and recent findings emphasize its ability to support processes like meaning-making, emotional responses, and mental imagery following psychedelic administration, ultimately contributing to positive clinical outcomes in psychedelic therapy [92].

## 3. 5-HT2A Receptor Signaling

Psychedelics have a similar chemical structure to serotonin (5-HT) [93]. Recent studies using advanced techniques like X-ray diffraction and cryo-electron microscopy have revealed the crystal structures of serotonin receptors when they are bound to psychedelic substances like LSD, psilocin, and others [62,94]. These studies provide new insights into how psychedelics work at the molecular level and their potential use in developing psychedelic-based treatments.

Phenethylamines exhibit a greater level of specificity for 5-HT2A, 5-HT2B, and 5-HT2C receptors in comparison to tryptamines and ergolines [3,95]. In the past, it was widely believed that the therapeutic effects of psychedelics were mainly attributed to their activation of the 5-HT2AR [96,97,98,99], but recent research has revealed the involvement of other receptors in producing these effects [51]. This belief is substantiated by evidence showing that the particular neural circuits in the central nervous system impacted by serotonergic psychedelics align with the distribution of serotonin receptors (Figure 2) [100]. 

The occupancy of 5-HT2ARs also exhibits a correlation with the subjective effects induced by psychedelics and these effects are suppressed by serotonin-2A antagonists [70,100,101,102,103,104,105]. The 5-HT2ARs are prominently found in pyramidal neurons situated in layer 5 of the neocortex, the thalamus, and the reticular nucleus, with a significant concentration observed in the prefrontal cortex (PFC) [104,106,107,108,109,110,111,112,113]. These brain areas are associated with functions like visual perception and attention, which likely contribute to the broad and diverse effects of psychedelics on cognitive, perceptual, and emotional processes. However, it is worth noting that other studies have indicated that the greatest concentration of these receptors is found in the striate and extrastriate visual cortex [80,104]. This observation could explain the frequent occurrence of visual hallucinations, a distinctive hallmark of classic psychedelics. Moreover, a recent study used a whole-brain model of serotonergic neuromodulation to investigate the entropic effects of 5-HT2AR activation, confirming the earlier findings of increased entropy and emphasizing that the most significant changes in entropy occurred in the visuo-occipital regions. Interestingly, this study also revealed that the overall reorganization of brain activity was more strongly associated with the brain’s anatomical connectivity than to the density of 5-HT2AR, offering valuable insights into the mechanisms behind the psychedelic experience and the broader regulation of brain functions by pharmacological means [114]. 

The expression of HT2AR is highest in excitatory neurons within the cortex, but it is also found in inhibitory interneurons. Specifically, in the prefrontal cortex, 5-HT2ARs are predominantly located on the postsynaptic side [115].

The 5-HT2AR, classified as a Class A G-protein-coupled receptor, is activated by its natural ligand, 5-HT, acting as an agonist. However, in contrast to 5-HT, psychedelic agonists targeting this receptor induce profound changes in perception and cognition. These effects can be explained by the ternary complex model of receptor activity for agonists [116,117]. This model suggests that drug molecules can shift the equilibrium between different receptor conformations, leading to “bias” in activating either G-protein-dependent or β-arrestin-dependent signaling pathways (Figure 3) [118,119,120]. In a recent study, cryo-EM structures of LSD-bound HTR2B provided snapshots of LSD’s action. These revealed transitions from transducer-free, partially active states to transducer-coupled, fully active states, highlighting the potential for biased agonists with functional selectivity, which could be safer and more effective drugs for specific G-protein-coupled receptors (GPCRs) [121].

In the context of Gq-dependent signaling, both the psychedelic and non-psychedelic activators of 5-HT2AR stimulate Gq-like G proteins [123]. The process begins with the activation of PLCγ (Figure 3). This activation subsequently initiates the release of intracellular calcium through inositol trisphosphate (IP3) and triggers the activation of protein kinase C (PKC) via diacylglycerol (DAG). This, in turn, sets off various interconnected downstream pathways, including but not limited to the ERK, CREB, and mTOR pathways (Figure 3). Ultimately, these cascading events result in neuronal firing through multiple mechanisms, such as membrane depolarization, reduced afterhyperpolarization, and decreased spike frequency adaptation [124,125,126,127] (Figure 3). The entry of calcium, along with the activation of calmodulin-dependent protein kinase II (CaMKII), which interacts with NMDA-type glutamate receptors and phosphorylates AMPA-type glutamate receptor subunits, leads to the specific strengthening and structural fortification of synapses, contributing to the process of long-term potentiation (LTP) that enhances synaptic connections during learning (Figure 3) [128,129]. Beyond G-protein-coupled pathways, 5-HT2AR agonism can also engage in β-arrestin signaling via PI3K and AKT (Figure 3) [96,130,131]. Both of these signaling pathways can contribute to neural plasticity [132,133], and may underlie the persistent improvements observed in psychiatric disorders. 

However, recent research conducted in mice suggests that the therapeutic effects of psychedelics may be most closely linked to the activation of the BNDF-TrkB signaling pathway (as discussed in Section 4), rather than the 5-HT2AR [51]. Specifically, several studies have shown that the “head twitch response” in rodents, which is used as a proxy for the hallucinogenic effects of psychedelics in humans [95], relies on 5-HT2AR activation [51,134,135], but that blocking the 5-HT2AR in mice did not prevent induced structural plasticity changes [51]. However, selectively restoring 5-HT2ARs in cortical pyramidal neurons was sufficient to rescue hallucinogen-induced head twitching in transgenic mice lacking 5-HT2Ars [136,137].

## 4. Classic Psychedelics Facilitate Plasticity via the TrkB-BDNF Signaling Pathway

Neuroplasticity likely plays a pivotal role in the therapeutic effects of psychedelics. Depression has been linked to reduced neurogenesis and neurotrophic activity [138]. Psychedelics, along with selective serotonin reuptake inhibitors (SSRIs) and certain non-hallucinogenic psychedelics, induce both structural and functional changes in cortical neurons, enhancing neurogenesis and thereby contributing to long-term benefits and improved stress-related behavior [67,139,140,141,142,143,144,145,146]. 

Similarly, BDNF plays a crucial role in neuronal growth and plasticity, and reduced levels have been linked to depression [147,148,149]. Conversely, activities like sleep and exercise have been shown to elevate BDNF levels [150,151], and greater baseline BDNF levels correspond to more significant improvements in depressive symptoms associated with antidepressant treatment [152,153]. Studies have shown that psychedelics, including LSD, ayahuasca, and psilocybin, can elevate BDNF levels, and this may contribute to their antidepressant effects [57,154,155,156].

Recent findings have shown that classic psychedelics like LSD bind to the TrkB receptor (Figure 3) and enhance BDNF signaling [51]. Other antidepressants, such as fluoxetine and ketamine, also bind to TrkB to facilitate neuroplasticity [157,158]. What is particularly noteworthy is that LSD’s affinity for TrkB has been shown to be up to 1000 times greater than that of other antidepressants [51,158]. This may contribute to the fast and potent induction of neuroplasticity and more persistent behavioral effects produced by psychedelics when compared with other antidepressants [159]. A specific point mutation in the transmembrane domain of TrkB that disrupts the binding of LSD to TrkB eliminates the induction of neuroplasticity and long-term plastic responses but has no impact on the head-twitch response. Furthermore, 5-HT2A receptor antagonists do not effectively stop the LSD-induced TrkB dimerization and the associated neurotrophic signaling, as well as the growth of spines and dendrites, along with the antidepressant-like behavioral effects. Taken together these findings suggest that these effects are separate and distinct from one another [51].

Psychedelics, like other antidepressants, do not directly activate TrkB receptors but rather facilitate their effects allosterically by enhancing the actions of extracellular BDNF released in active synapses (Figure 3) [51,158]. This initiates a sequence of intracellular transduction cascades that ultimately facilitate plasticity, as illustrated in Figure 3. Additionally, psychedelics promote neuroplasticity by increasing the trafficking of AMPA-type glutamate receptors and elevating BDNF levels [51] (Figure 3) thus promoting the maintenance and strengthening of activity-dependent plasticity in active synapses and fostering long-term brain modifications [160].

Acute psychedelic administration in rodents has been shown to increase the expression of immediate early genes associated with plasticity and learning, underscoring the profound molecular and neuronal effects of these compounds [59,161]. Furthermore, single doses of psychedelics can induce lasting epigenetic changes, particularly in enhancer regions linked to synaptic assembly, persisting for days after their use [59]. 

However, the antidepressant effects of BDNF and TrkB signaling in promoting plasticity must be understood within the context of the neuronal/circuit and brain level (Figure 1). Stress triggers specific changes in the adult brain’s reward circuitry, leading to the increased expression of BDNF and TrkB in the nucleus accumbens (NAc) and ventral tegmental area (VTA), while reducing expression in other stress-related brain regions like the hippocampus [162]. Repeated exposure to social defeat stress in mice elevates BDNF levels in the Nac [163], which is associated with heightened susceptibility to mood disorders like anxiety and depression [164]. Social aversion also involves BDNF secretion and contributes to neuroadaptation in the mesocorticolimbic system [165,166]. BDNF-containing pathways from the VTA and PFC to the Nac play a role in this process. Inhibiting TrkB receptors in the Nac shell prevents the stress-induced inhibition of TrkB in both the Nac core and shell [167], highlighting the crucial role of TrkB receptors in maintaining homeostasis within the reward circuitry.

Aversive motivation, such as anxiety and depression, is considered a key factor in driving substance abuse through negative reinforcement (temporary relief) [168]. A recent review [166] discusses the role of BDNF in aversive motivation and its differential effect on the mesolimbic system as compared to the hippocampus and frontal cortex. Reduced BDNF levels in the hippocampus and prefrontal cortex have been linked to aversive motivation [166,169], while increased BDNF levels are found in the mesolimbic system in conditions such as drug addiction [166], epilepsy [170], and neuropathic pain [171]. Therefore, a therapy focusing on both decreasing BDNF levels in the mesolimbic system and increasing BDNF levels in the hippocampus and frontal cortex could effectively treat pathological aversive motivation [166,172]. Classical psychedelics seem to fulfill this dual function. Psilocybin, for example, enhances neural plasticity in the hippocampus and frontal cortex, aiding in the alleviation of aversive motivation associated with fear conditioning in animal models, while reducing BDNF-related increased plasticity in the mesolimbic system [142,144,166]. BDNF triggers a shift in the function of VTA receptors for gamma-aminobutyric acid type A (GABA-A) on GABA neurons, changing inhibitory effects to excitatory [173]. Classical hallucinogens, particularly 5-HT2A agonists, are proposed to revert the drug-dependence associated with substance abuse disorder. This is achieved by reducing BDNF-induced plasticity in VTA GABAergic neurons, leading to the release of inhibitory dopaminergic signaling in the mesolimbic system, subsequently alleviating the negative symptoms associated with aversive motivation [166]. The long-term effects of 5-HT2A agonists on NMDA-like glutamate receptors are suggested, but further experimental validation is needed [166,174]. It is worth emphasizing that the exact molecular mechanisms responsible for this differential impact of the BDNF-TrkB pathway on neuronal plasticity and stress remain unknown. Both the timing of synaptic activity and the various temporal phases of synaptic enhancement have been suggested as important factors in determining the neurotrophin dependence of plasticity in the hippocampus [175]. Additionally, the regulation of TrkB cell surface expression and trafficking has been proposed as mechanisms that can alter the responsiveness of target cells to BDNF, potentially leading to abnormal synaptic plasticity and disrupted network communication. These factors may be involved in various neuropsychiatric diseases [176].

As previously discussed, there has been a greater focus on understanding the mechanisms related to receptor localization in various brain regions and the effects of psychedelics, particularly in the context of 5-HT2A receptors. However, there are still gaps in our knowledge in this area. Psychedelics activate 5-HT2A receptors, which are found post-synaptically on layer 5 and 6 pyramidal neurons and GABAergic interneurons [106]. As mentioned previously, inhibitory GABAergic interneurons in the cortex and subcortical structures also express 5-HT2A receptors [109]. The overall effect appears to be the excitation of layer 5 pyramidal neurons and an increase in extracellular glutamate levels, leading to greater stimulation of AMPA receptors [106,177]. In vivo experiments, DOI, for example, demonstrated a pronounced overall excitatory effect on pyramidal neurons in the rat prefrontal cortex, with some inhibition of GABAergic interneurons [178]. In a different study, lower doses of DOI resulted in the activation of neuronal populations in the rat orbitofrontal cortex and anterior cingulate cortex, while higher doses tended to suppress activity in these regions, possibly influenced by receptor density [115,179]. Hence, it seems that psychedelics can have different modulatory effects across cortical regions, depending on factors such as dosage, the specific drug used, and the density of 5-HT2A receptors in various neuronal populations [106]. 

Additionally, the precise molecular pathways that may modify neuroplasticity after 5-HT2A receptor stimulation remain incompletely understood (Figure 3). A prevailing theory suggests that AMPA receptor activation initiates a positive feedback loop, with enhanced AMPA receptor activity leading to increased BDNF secretion, which in turn stimulates TrkB receptors and mTOR [67]. This cascade of events sustains BDNF production and prolonged AMPA and mTOR activation, which seems necessary for the enhanced dendritic growth observed after psychedelic stimulation [180]. These effects likely occur in synapses and circuits expressing 5-HT2A receptors since BDNF primarily acts locally and does not diffuse extensively after release [181]. Hence, this theory proposes that the neuroplasticity linked to TrkB-BDNF signaling arises as a consequence of downstream effects resulting from 5-HT2A receptor signaling and TrkB activation. This contradicts the findings we have discussed [51,158], suggesting that TrkB signaling represents an independent and distinct mechanism of plasticity. Therefore, further research is required to address these aspects.

## 5. What Do 5-HT2A and TrkB Pathways Reveal about the Role of Subjective Experience in Serotonergic Psychedelic Therapy?

The acute hallucinogenic effects of psychedelics have limited their widespread clinical use, necessitating specialized medical supervision in controlled clinical settings for extended sessions [182,183,184]. 

Concerns about psychedelics inducing psychosis can be traced back to the time when LSD was banned, with a focus on instances of “acid casualties” that significantly influenced the way society viewed psychedelics. It is worth noting that such cases are infrequent, especially in clinical applications [185]. However, in individuals who have a genetic predisposition, which can be indicated by a family history of psychiatric disorders like schizophrenia or bipolar disorder, there is a potential risk that psychedelics could trigger psychotic episodes [186,187,188,189,190,191]. This risk is likely associated with alterations at the molecular level in brain regions responsible for perception, cognition, and mood regulation. As a precaution, individuals with a family history of schizophrenia or bipolar disorder, who may have a genetic susceptibility to psychotic illnesses, are generally excluded from clinical treatments involving psychedelics [191]. With rigorous screening procedures in place, there have been no reported incidents of psychotic episodes in modern clinical trials involving psychedelics [185]. Zeifmann and colleagues [192] conducted a systematic review on classic psychedelics and suicide risk. Their findings indicate that psychedelic therapy can potentially decrease suicidality in specific clinical psychiatric groups, while classic psychedelic use might offer some protection against suicidal tendencies [192]. However, unsupervised and uncontrolled psychedelic use can, in rare instances, lead to fatal outcomes, including suicide. The authors emphasize that their conclusions are based on a limited dataset, mainly comprising case studies and small retrospective reports [185,192].

Another risk associated with the hallucinogenic effects of psychedelics is the occurrence of flashbacks and Hallucinogen Persisting Perception Disorder (HPPD), in which certain individuals may experience unwanted and distressing visual disturbances that often resemble the effects of psychedelics, even after the drug has cleared from their system [193]. The molecular mechanisms behind HPPD are not fully understood, but it may involve alterations in visual processing and neurotransmitter systems. The DSM-V reports a broad range of prevalence rates for Hallucinogen Persisting Perception Disorder (HPPD), with figures as high as 4.2% based on an online questionnaire [193], while other studies have reported rates as low as 1 in 50,000 [194]. Despite involving thousands of participants in psychedelic research and clinical trials spanning the past two decades, there have been no documented instances of HPPD [44,185,195]. Clinical settings report lower HPPD incidence, likely due to screening and preparation [191]. While some studies indicate no identifiable risk factors [196], others suggest a link between HPPD and anxiety or panic reactions during the initial experience, potentially resembling a trauma response or health anxiety [197].The activation of serotonin 5-HT2A receptors, along with the presence of oxidative stress and the occurrence of apoptotic cell death, has been proposed as potential mechanisms that might underlie neurotoxicity resulting from the use of novel psychoactive substances (NPS) with psychedelic properties. Nevertheless, it remains uncertain to what degree these mechanisms actually play a role in the adverse effects observed in humans following the consumption of stimulants and psychedelics NPS [198].

Due to the hallucinogenic effects, individuals with a family history of bipolar disorder or schizophrenia have been excluded from participating in clinical trials involving psychedelics for depression, despite the potential therapeutic benefits that could be valuable for these populations. [183,184,191]. 

However, recent findings suggest that the TrkB-dependent effects of psychedelics on plasticity may be separated from their hallucinogenic-like effects mediated by 5-HT2A receptors [51,199,200]. This suggests the potential to discover compounds or treatment combinations that retain some of the antidepressant effects of psychedelics without the hallucinogenic effects [62,201]. Some of these compounds include isoDMT [202], tabernanthalog [199], AAZ-A-154 [203], and 2-bromo-LSD [204], achieved by modifying the structures of known hallucinogenic compounds. The availability of high-resolution structures of 5-HT receptors in complex with psychedelics promises to expedite the search for novel psychedelic and non-hallucinogenic 5-HT2A agonists [62,94,205]. The emerging data on the therapeutic role of psychedelic binding to TrkB receptors, enhancing neuroplasticity, will also drive computational analysis in this new direction.

A significant question is whether the psychedelic experience is an essential aspect of the therapeutic effect of serotonergic psychedelics [206,207]. One hypothesis is that the psychedelic experience is not strictly necessary for therapeutic effects [208]. This was indicated by a recent study in mice, which showed TrkB binding independent of 5-HT2AR activation [51], and is also consistent with the reports of microdosers who frequently claim clinically relevant benefits without marked subjective effects [209]. Another hypothesis is that hallucinogenic experiences grant access to underlying psychological and emotional phenomena that can be addressed, modified, and reinforced through psychotherapy [210]. Supporting this hypothesis, the reduction in prefrontal 5-HT2AR activity resulting from classical hallucinogens could explain certain aspects of their impact on depression and anxiety [211]. This suggests that the alterations in 5-HT2AR activity associated with hallucinogenic effects may play a crucial role in the therapeutic aspects of these substances.

Importantly, species differences can influence the properties of 5-HT2ARs, and it may turn out that animal models are unable to tell us much about the importance of psychedelic experiences for clinical outcomes in humans [212]. At present, there is insufficient evidence to conclusively support either hypothesis, and further research is needed to empirically address this question [207].

## 6. Glutamate Signaling: A Shared Regulator of Neuroplasticity in Hallucinogens and Dissociative Anesthetics

Glutamate signaling deficiencies are a significant characteristic of major depressive disorders [213]. Psychedelics promote neuroplasticity by increasing extracellular glutamate levels in the prefrontal cortex [177,214,215] and triggering the release of neurotrophic factors like BDNF that enhance neural plasticity (Figure 3) [56,57,155]. This process involves the activation of calcium/calmodulin-dependent protein kinase II (CaMKII) and activation of metabotropic glutamate receptors (mGluR), including mGluR2 and mGluR3, which influence G-protein coupling and downstream signaling pathways linked to rapid antidepressant effects (Figure 3) [129,216,217,218,219]. Classic psychedelics induce glutamate release in the medial prefrontal cortex, leading to the sustained activation of AMPA receptors and subsequent BDNF release, which in turn activates mTOR signaling and upregulates neuroplasticity-related genes and synaptic protein synthesis, ultimately enhancing social behavior in mice (Figure 3) [22,52,220,221]. These effects can be blocked by specific 5-HT2AR antagonists, AMPAR antagonists, mGluR2 positive allosteric modulators, and NMDAR antagonists [179,222,223,224]. On the other hand, DOI and LSD elevate glutamate and BDNF levels and enhance NMDAR-mediated transmission [177,215,225]. These findings suggest that psychedelics may promote neuroplasticity through glutamate-driven AMPAR activation, with potential therapeutic implications for mood disorders and stress-related learning [226,227].

Dissociative anesthetics like ketamine block NMDARs, leading to increased glutamate release in the medial prefrontal cortex (Figure 2) [214,228,229,230,231]. The elevated extracellular glutamate levels contribute to the psychotropic effects of ketamine and PCP, and these effects can be influenced by AMPAR antagonists or mGluR2 and mGluR3 agonists [229,232]. Similar to classic psychedelics, ketamine also promotes neuronal structural remodeling and plasticity, suggesting a common mechanism for neuroplasticity shared between them [52,233,234]. Both classes of drugs stimulate AMPARs by increasing extracellular glutamate levels and elevate BDNF levels in brain areas implicated in depression [22,145,220,222,224,235,236,237,238], facilitating the adaptive rewiring of pathological neurocircuitry and explaining their sustained therapeutic effects [52].

## 7. Additional Receptors and Pathways Contributing to the Mechanisms of Action of Psychedelics

Psychedelics have a complex pharmacology, interacting with various serotonin 5-HT receptors, such as 5-HT1, 5-HT4, 5-HT5, 5-HT6, and 5-HT7, in addition to the well-known high affinity for 5-HT2 receptors [239,240]. LSD, for example, has a high affinity for multiple human 5-HT receptors, as well as D1, D2, D3, and D4 dopamine and α1 and α2-adrenergic receptors [94,241,242]. Ergolines also exhibit significant intrinsic activity at both dopamine D2 receptors and α-adrenergic receptors [243]. Psilocin also strongly activates several 5-HT receptors including 5-HT1D, 5-HT2A, 5-HT2B, 5-HT2C, 5-HT5, 5-HT6, and 5-HT7 [244,245], histamine-1, α2A, and α2B adrenergic receptors, and dopamine D3 receptors [50].

Some psychedelics like tryptamines can bind to 5-HT1A receptors and decrease neuronal excitability [127,246]. LSD can also impact 5-HT1A receptors by desensitizing them, increasing serotonin release [247]. Additionally, regions abundant in 5-HT1A receptors, especially limbic areas, have shown decreased signal intensity and inherent connectivity in whole-brain modeling studies [100]. The distribution and abundance of these receptor subtypes determine how psychedelics affect neuronal activity [248]. 

The dopaminergic system (Figure 2) becomes active when exposed to psilocybin, resulting in increased dopamine levels that correlate with feelings of euphoria and depersonalization [249]. Psilocybin does not directly interact with dopamine receptors; instead, it may elevate striatal dopamine levels through the activation of 5-HT1A receptors [250]. However, when dopamine D2 receptors are blocked with haloperidol, these effects are only moderately diminished and selectively impact positive derealization, with no influence on visual hallucinations or working memory, suggesting a relatively modest involvement of the dopaminergic system in these psilocybin responses [102]. Unlike psilocybin, LSD does exhibit significant intrinsic activity at dopamine D2 receptors [50,240], which is characterized by a phased response, initially activating 5-HT2ARs and later D2 receptors [251]. Recent findings suggest that the subjective effects of 30 mg psilocybin and 100 or 200 mg of LSD, despite their varied pharmacological profiles and receptor affinities, cannot be reliably distinguished, challenging the idea of a precise link between receptor activation and phenomenological experiences [57].

Lately, there has been growing interest in the potential use of psychedelics, which have a broad impact on 5-HT2 receptors, in the treatment of Substance Use Disorder (SUD), with a particular focus on the involvement of the 5-HT2C receptor [252].

Another receptor that may contribute to the therapeutic effects of psychedelics is the trace amine-associated receptor (TAAR), as psychedelics activate the TAAR1 receptor, subsequently exerting an inhibitory influence on dopaminergic activity [253,254].

The Sigma-1 receptor (Sig-1R) is a small integral membrane protein that localizes at the interface of the endoplasmic reticulum (ER) and mitochondria, known as the mitochondria-associated ER membrane (MAM) (Figure 3) [255,256]. These contact sites play a pivotal role in various cellular processes, including calcium regulation, lipid metabolism, and apoptosis [256]. Consequently, Sig-1R has been implicated in promoting cell survival, neuroprotection, neuroplasticity, and neuroimmunomodulation [257].

Recent studies exploring the interactions between psychedelics and Sig-1Rs have opened up a new avenue of research regarding their potential impact on mitochondrial function and the associated MAMs [258]. Much like 5-HTRs, Sig-1Rs can influence the activity of Ca^2+^ channels, either enhancing or inhibiting them, consequently regulating intracellular Ca^2+^ levels, which can affect lipid transfer between the ER and mitochondria [259]. 

The chaperone function of the Sig-1R is essential for regulating various cellular processes [260]. It interacts with proteins within the endoplasmic reticulum, ensuring their correct folding and appropriate functioning. This role is crucial for maintaining neuronal homeostasis and protecting against cellular stress [256,257,260] and may extend beyond the central nervous system to encompass a broader role in cellular protective mechanisms [261].

DMT has been identified as a natural, endogenous ligand for Sig-1R [257,262]. This interaction can influence mitochondrial energy production and respiration rates. At its endogenous affinity concentrations (14 μM), DMT binds to Sig-1R, leading to the dissociation of Sig-1R from the immunoglobulin protein (BiP). This dissociation enables Sig-1R to function as a molecular chaperone for IP3R [263]. This leads to enhanced Ca^2+^ signaling and a substantial increase in ATP production (Figure 3) [264]. However, at higher concentrations (100 μM), DMT induces the translocation of Sig-1Rs from the MAM to the plasma membrane, leading to the inhibition of ion channels [262,263,265] (Figure 3). In both human in vitro and animal in vivo studies, DMT has demonstrated potent neuroprotective and neurogenerative effects via Sig1R [266,267].

Furthermore, psychedelics binding to Sig-1R may have an impact on the production of reactive oxygen species (ROS), which are natural byproducts of mitochondrial metabolism. During glutamate excitotoxicity, excessive stimulation of glutamate receptors can lead to uncontrolled propagation of action potentials through neurons, triggering the influx of Ca^2+^ ions into the cytoplasm. This influx may induce mitochondrial respiration and the release of ROS [268]. Sig1R agonists such as dimemorfan and dipentylammonium have been shown to mediate neuroprotection against glutamate excitotoxicity [269,270], and a reduction in glutamate toxicity in vivo has also been suggested for DMT [271,272].

Research into how psychedelics affect Sig-1R and their consequences for mitochondrial function and MAMs is a promising but emerging field, with a need for further investigation to uncover precise mechanisms and potential therapeutic applications.

## 8. The Potential of Psychedelics as Anti-Inflammatory Agents

Psychedelics have shown promise in reducing inflammatory markers associated with depression, addiction, and anxiety [58,273,274]. They can inhibit cytokines and several studies have reported decreases in inflammatory markers in human cells following the administration of various psychedelics [274,275,276,277,278,279,280,281,282,283,284]. Furthermore, psychedelics trigger a biochemical stress response, releasing catecholamines and glucocorticoids, and this, counterintuitively, may contribute to their transformative effects [247,285,286,287]. Although the mechanism underlying this response is not fully understood, it may involve direct actions on 5-HT2ARs in the hypothalamus and microglia [288,289,290,291]. Most immune cells possess 5-HT receptors, leading to the hypothesis that serotonergic psychedelics affect immunomodulatory agents by activating 5-HT2AR agonism [292]. An alternative hypothesis suggests that the altered state of consciousness induced by psychedelics may activate the acute stress response, potentially contributing to their neuroplastic effects, although the role of the stressful aspects of the psychedelic experience in therapeutic benefits is a subject of debate [293,294].

Furthermore, the Sig-1R’s expression is not limited to specific regions within the CNS, but it is also found in immune cells [295]. Sig-1R ligands have been shown to exhibit potent immunoregulatory properties as evidenced by their ability to elevate the secretion of the anti-inflammatory cytokine IL-10 [261,296,297,298,299], by regulating the activation of the transcription factors nuclear factor kappa B (NF-кB) and several MAPKs [269].

Although research into the anti-inflammatory effects of psychedelics is at an early stage, this line of research shows potential for treating neurological disorders marked by chronic immune activity such as Alzheimer’s and Parkinson’s, as well as autoimmune-related conditions [36].

## 9. Conclusions

The cellular neurobiology of psychedelics is a complex and multifaceted field of study that holds great promise for understanding the mechanisms underlying their therapeutic effects. These substances engage intricate molecular/cellular, circuit/network, and overall brain-level mechanisms, impacting a wide range of neurotransmitter systems, receptors, and signaling pathways. This comprehensive review has shed light on the mechanisms underlying the action of psychedelics, particularly focusing on their activity on 5-HT2A, TrkB, and Sig-1A receptors. The activation of 5-HT2A receptors, while central to the psychedelic experience, is not be the sole driver of their therapeutic effects. Recent research suggests that the TrkB-BDNF signaling pathway may play a pivotal role, particularly in promoting neuroplasticity, which is essential for treating conditions like depression. This delineation between the hallucinogenic and non-hallucinogenic effects of psychedelics opens avenues for developing compounds with antidepressant properties and reduced hallucinogenic potential. Moreover, the interactions between psychedelics and Sig-1Rs have unveiled a new avenue of research regarding their impact on mitochondrial function, neuroprotection, and neurogeneration.

Overall, while our understanding of the mechanisms of psychedelics has grown significantly, there is still much research needed to unlock the full potential of these compounds for therapeutic purposes. Further investigation into their precise mechanisms and potential clinical applications is essential in the pursuit of new treatments for various neuropsychiatric and neuroinflammatory disorders.

## Figures and Tables

**Figure 1 biology-12-01380-f001:**
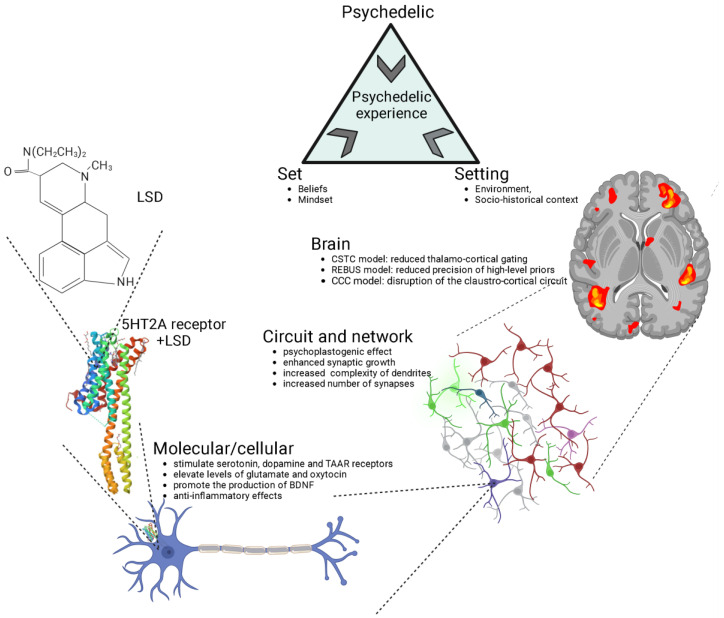
Psychedelics exert their effects through various levels of analysis, including the molecular/cellular, the circuit/network, and the overall brain. The crystal structure of serotonin 2A receptor in complex with LSD is sourced from the RCSB Protein Data Bank (RCSB PDB) [62]. LSD, lysergic acid diethylamide; 5-HT2A, serotonin 2A; CSTC, cortico-striato-thalamo-cortical [63]; REBUS, relaxed beliefs under psychedelics model [64]; CCC, claustro-cortical circuit [65]. Generated using Biorender, https://biorender.com/, accessed on 4 September 2023.

**Figure 2 biology-12-01380-f002:**
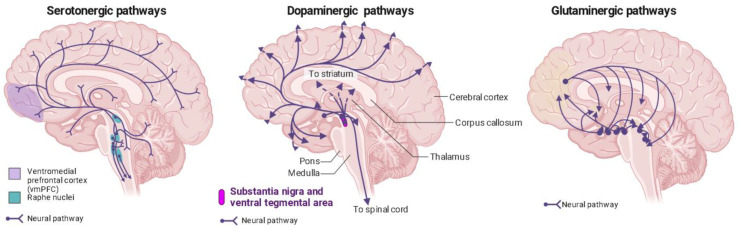
Distribution of serotonin, dopamine, and glutaminergic pathways in the human brain. Ventromedial prefrontal cortex (vmPFC) in purple; raphe nuclei in blue. Generated using Biorender, https://biorender.com/, accessed on 4 September 2023.

**Figure 3 biology-12-01380-f003:**
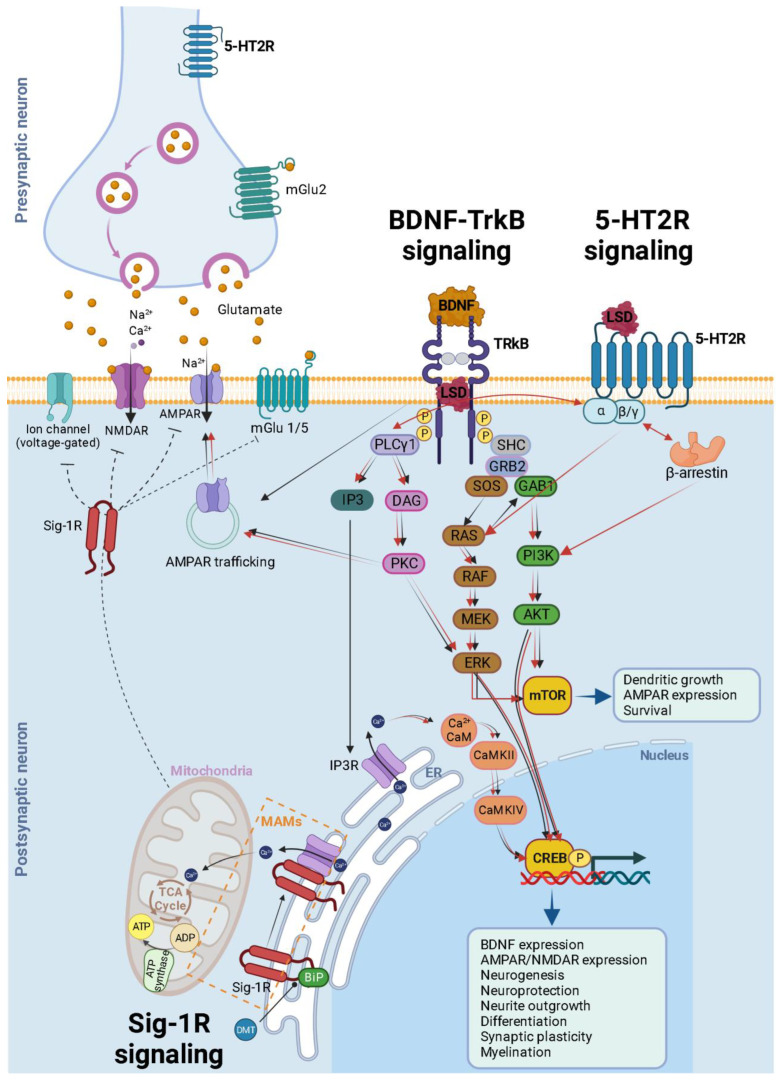
Schematic and simplified overview of the intracellular transduction cascades induced by 5-HT2AR TrkB and Sig-1R receptor activation by psychedelics. It is essential to emphasize that our understanding of the activation or inhibition of specific pathways and the precise molecular mechanisms responsible for triggering plasticity in specific neuron types remains incomplete. This figure illustrates the mechanisms associated with heightened plasticity within these pathways. Psychedelics (such as LSD, psilocin, and mescaline) bind to TrkB dimers, stabilizing their conformation. Furthermore, they enhance the localization of TrkB dimers within lipid rafts, thereby extending their signaling via PLCγ1. The BDNF/TrkB signaling pathway (black arrows) initiates with BDNF activating TrkB, prompting autophosphorylation of tyrosine residues within TrkB’s intracellular C-terminal domain (specifically Tyr490 and Tyr515), followed by the recruitment of SHC. This, in turn, leads to the binding of GRB2, which subsequently associates with SOS and GTPase RAS to form a complex, thereby initiating the ERK cascade. This cascade ultimately results in the activation of the CREB transcription factor. CREB, in turn, mediates the transcription of genes essential for neuronal survival, differentiation, BDNF production, neurogenesis, neuroprotection, neurite outgrowth, synaptic plasticity, and myelination. Activation of Tyr515 in TrkB also activates the PI3K signaling pathway through GAB1 and the SHC/GRB2/SOS complex, subsequently leading to the activation of protein kinase AKT and CREB. Both Akt and ERK activate mTOR, which is associated with downstream processes involving dendritic growth, AMPAR expression, and overall neuronal survival. Additionally, the phosphorylation of TrkB’s Tyr816 residue activates the phospholipase Cγ (PLCγ) pathway, generating IP3 and DAG. IP3 activates its receptor (IP3R) in the endoplasmic reticulum (ER), causing the release of calcium (Ca^2+^) from the ER and activating Ca^2+^/CaM/CaMKII which in turn activates CREB. DAG activates PKC, leading to ERK activation and synaptic plasticity. After being released into the extracellular space, glutamate binds to ionotropic glutamate receptors, including NMDA receptors (NMDARs) and AMPA receptors (AMPARs), as well as metabotropic glutamate receptors (mGluR1 to mGluR8), located on the membranes of both postsynaptic and presynaptic neurons. Upon binding, these receptors initiate various responses, such as membrane depolarization, activation of intracellular messenger cascades, modulation of local protein synthesis, and ultimately, gene expression. The surface expression and function of NMDARs and AMPARs are dynamically regulated through processes involving protein synthesis, degradation, and receptor trafficking between the postsynaptic membrane and endosomes. This insertion and removal of postsynaptic receptors provides a mechanism for the long-term modulation of synaptic strength [122]. Psychedelic compounds exhibit a high affinity for 5-HT2R, leading to the activation of G-protein and β-arrestin signaling pathways (red arrows). Downstream for 5-HT2R activation, these pathways intersect with both PI3K/Akt and ERK kinases, similar to the BDNF/TrkB signaling pathway. This activation results in enhanced neural plasticity. A theoretical model illustrating the signaling pathway of DMT through Sig-1R at MAMs suggests that, at endogenous affinity concentrations (14 μM), DMT binds to Sig-1R, triggering the dissociation of Sig-1R from BiP. This enables Sig-1R to function as a molecular chaperone for IP3R, resulting in an increased flow of Ca^2+^ from the ER into the mitochondria. This, in turn, activates the TCA cycle and enhances the production of ATP. However, at higher concentrations (100 μM), DMT induces the translocation of Sig-1Rs from the MAM to the plasma membrane (dashed inhibitory lines), leading to the inhibition of ion channels. BDNF = brain-derived neurotrophic factor; TrkB = tropomyosin-related kinase B; LSD = lysergic acid diethylamide; SHC = src homology domain containing; SOS = son of sevenless; Ras = GTP binding protein; Raf = Ras associated factor; MEK = MAP/Erk kinase; mTOR = mammalian target of rapamycin; ERK = extracellular signal regulated kinase; GRB2 = growth factor receptor bound protein 2; GAB1 = GRB-associated binder 1; PLC = phospholipase C γ; IP3 = inositol-1, 4, 5-triphosphate; DAG = diacylglycerol; PI3K = phosphatidylinositol 3-kinase; CaMKII = calcium/calmodulin-dependent kinase; CREB = cAMP-calcium response element binding protein; AMPA = α-amino-3-hydroxy-5-methyl-4-isoxazolepropionic acid; Sig-1R = sigma-1 receptor; DMT = N,N-dimethyltryptamine; BiP = immunoglobulin protein; MAMs = mitochondria-associated ER membrane; ER = endoplasmic reticulum; TCA = tricarboxylic acid; AT P = adenosine triphosphate; ADP = adenosine diphosphate. Generated using Biorender, https://biorender.com/, accessed on 20 September 2023.

## Data Availability

Data sharing not applicable.

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
