# Peer review of "A Comprehensive Review of the Current Status of the Cellular Neurobiology of Psychedelics"

_biology, 2023, doi:10.3390/biology12111380_

Round 1

Reviewer 1 Report

Comments and Suggestions for Authors

The authors present a critical perspective on the cellular neurobiology of psychedelics and emphasize the imperative for further exploration in this domain. They also highlight several areas of opportunity, including the need for additional research, ethical considerations, and broadening the scope of study. The article offers novel albeit incomplete insights into recent advancements in understanding psychedelics as potential therapeutic agents and some of their cellular pathways. However, to avoid being perceived as a mere endorsement of psychedelic studies, the authors should delve deeper into the potential negative effects and risks associated with the therapeutic use of these substances, while discussing their neurophysiological and/or molecular underpinnings.

For a comprehensive review, the authors should scrutinize studies linking BDNF to aversive motivation, particularly in the mesolimbic system and, more precisely, in GABAergic interneurons. BDNF and its plasticity in GABAergic interneurons have been associated with various pathologies such as epileptic seizures, neuropathic pain, depression, and addictions. Indeed, the potential of psychedelics lies in their ability to act as a sort of "magic bullet," reducing plasticity in GABAergic interneurons while increasing it in pyramidal neurons. This topic warrants extensive exploration and discussion to provide a broader perspective.

The authors mention other receptor types beyond 5-HT2R, but, apart from 5HT1R and 5HT7R, they overlook one of the most important receptors, namely, the sigma1 receptor (not to mention its chaperone function). In doing so, the authors omit studies linking psychedelics to their potential impact on mitochondrial function. In this emerging mitochondriocentric era, the authors should not neglect the potential action of psychedelics on mitochondrial function and their associated membranes (MAMs).

Concerning the ancestral use of psychedelics, the authors overlook exposure to endogenous psychedelics and altered states of consciousness induced by other elements such as electromagnetism, extreme stress, and attention modification through various means. How does this differ from or resemble the consumption of psychedelics?

While this is a commendable review with up-to-date information, it remains partial and requires supplementation with the suggested information.

Author Response

We thank the Editor and Reviewers for careful review which helped improve the manuscript.

We have addressed and incorporated suggestions from both reviewers. Please find our answer to comments below as well as suggested text changes in the manuscript file with track changes.

Response to Reviewers comments

Reviewers' comments 1:

 The authors present a critical perspective on the cellular neurobiology of psychedelics and emphasize the imperative for further exploration in this domain. They also highlight several areas of opportunity, including the need for additional research, ethical considerations, and broadening the scope of study. The article offers novel albeit incomplete insights into recent advancements in understanding psychedelics as potential therapeutic agents and some of their cellular pathways. However, to avoid being perceived as a mere endorsement of psychedelic studies, the authors should delve deeper into the potential negative effects and risks associated with the therapeutic use of these substances, while discussing their neurophysiological and/or molecular underpinnings.

 We thank the reviewer for their positive feedback on our review and for providing us with the opportunity to offer a more comprehensive perspective in this emerging field of research. In response to the reviewer's suggestion, we have indeed delved deeper into the potential negative effects and associated risks associated with the therapeutic use of psychedelics. It's important to note that our understanding of the molecular mechanisms underpinning these effects remains limited, primarily due to their infrequent occurrence and the nascent nature of this field. The scarcity of data may account for why our review, which primarily focuses on the molecular mechanisms of these substances, might have been perceived as an endorsement of psychedelics, considering the substantial evidence supporting their therapeutic benefits and the intricate molecular mechanisms linked to their positive effects. Nevertheless, we align with the reviewer's perspective and have made substantial additions to our content to ensure clarity on this matter and to avoid any potential misinterpretations. Please refer to paragraph 5 (lines 422-463), for further details.

 For a comprehensive review, the authors should scrutinize studies linking BDNF to aversive motivation, particularly in the mesolimbic system and, more precisely, in GABAergic interneurons. BDNF and its plasticity in GABAergic interneurons have been associated with various pathologies such as epileptic seizures, neuropathic pain, depression, and addictions. Indeed, the potential of psychedelics lies in their ability to act as a sort of "magic bullet," reducing plasticity in GABAergic interneurons while increasing it in pyramidal neurons. This topic warrants extensive exploration and discussion to provide a broader perspective.

 We thank the reviewers for this valuable suggestion. We have incorporated a new paragraph that delves into the relationship between BDNF and aversive motivation, with a specific focus on its role within the mesolimbic system. Please refer to paragraph 4 (lines 396-415).

The authors mention other receptor types beyond 5-HT2R, but, apart from 5HT1R and 5HT7R, they overlook one of the most important receptors, namely, the sigma1 receptor (not to mention its chaperone function). In doing so, the authors omit studies linking psychedelics to their potential impact on mitochondrial function. In this emerging mitochondriocentric era, the authors should not neglect the potential action of psychedelics on mitochondrial function and their associated membranes (MAMs).

 We thank the reviewer for this suggestion, and we have included this information in the manuscript. Please refer to paragraph 7 (lines 555-587) and paragraph 8 (lines 606-610).  

 Concerning the ancestral use of psychedelics, the authors overlook exposure to endogenous psychedelics and altered states of consciousness induced by other elements such as electromagnetism, extreme stress, and attention modification through various means. How does this differ from or resemble the consumption of psychedelics?

 While the primary focus of this paper revolves around the consumption of exogenous psychedelics, it is important to acknowledge that altered states of consciousness can also be achieved through non-pharmacological means such as electromagnetism, extreme stress, and attention modification, leading to changes in perception, cognition, and awareness. The key differences between these and the consumption of exogenous psychedelics include the origin and specificity of the compounds involved, the predictability of the experiences, and the potential for therapeutic use. Moreover, these methods may function through distinct yet potentially converging mechanisms – electromagnetic fields can influence brain activity, extreme stress can induce stress responses and altered consciousness, and attention modification techniques can impact perception and cognition. It is worth noting that despite these differences, there may be shared subjective experiences and potential therapeutic benefits among these various approaches to altered states of consciousness. Research into these diverse methods and their comparative effects on consciousness represents an exciting and ongoing area of study. Nevertheless, it's important to acknowledge that a comprehensive discussion of these mechanisms goes beyond the scope of this review.

 While this is a commendable review with up-to-date information, it remains partial and requires supplementation with the suggested information.

 We would like to express our gratitude to the reviewer for their valuable suggestions. The manuscript has been revised to incorporate the recommended information, significantly enhancing its overall quality.

Reviewer 2 Report

Comments and Suggestions for Authors

This review provides a comprehensive overview of the current state of research into the cellular neurobiology of psychedelics. It explores the potential therapeutic applications of these substances in addressing mental illness and emphasizes their pharmacological and molecular mechanisms, including the role of the 5-HT2A receptor and the TrkB-BDNF pathway. Overall, the paper is well-written and addresses an important and timely topic. I have a few suggestions/comments for the authors’ consideration to further improve this submission. 

#1. The number of references synthesized in this review is a testament to its comprehensiveness. While it could be helpful to include a summary of the literature search and review process in addition to the contextual synthesis. 

#2. The newly released FDA Guidance on Clinical Trials with Psychedelic Drugs is a timely update that can be incorporated into the Introduction section: https://www.fda.gov/news-events/press-announcemlients/fda-issues-first-draft-guidance-clinical-trials-psychedelic-drugs

#3. Line 167 - The authors mentioned that “The models and levels of analysis presented in Figure 1 can potentially be expanded to include extra-pharmacological factors…” Is there any potential to incorporate these factors into the current Figure 1? 

#4. Figure 2: Confirm the color code for “Ventromedial prefrontal cortex (vmPFC)” and “Raphe nuclei”. 

#5. Minor corrections are needed for the following sentences if applicable: 

  1. )Line 123: “The structural and neuroplastic changes brought about by…”

  2. )Line 209: “Notably,, ...”

Author Response

Reviewers' comments 2:

 This review provides a comprehensive overview of the current state of research into the cellular neurobiology of psychedelics. It explores the potential therapeutic applications of these substances in addressing mental illness and emphasizes their pharmacological and molecular mechanisms, including the role of the 5-HT2A receptor and the TrkB-BDNF pathway. Overall, the paper is well-written and addresses an important and timely topic. I have a few suggestions/comments for the authors’ consideration to further improve this submission.

 We appreciate the reviewer's positive feedback on our review and their valuable suggestions and comments for its improvement.

 #1. The number of references synthesized in this review is a testament to its comprehensiveness. While it could be helpful to include a summary of the literature search and review process in addition to the contextual synthesis.

  We appreciate the reviewer's input. It's worth noting that this review follows a narrative format rather than a systematic one, making a detailed summary of the literature search less suitable for this particular review type.

 #2. The newly released FDA Guidance on Clinical Trials with Psychedelic Drugs is a timely update that can be incorporated into the Introduction section: https://www.fda.gov/news-events/press-announcemlients/fda-issues-first-draft-guidance-clinical-trials-psychedelic-drugs.

  We thank the reviewer for this suggestion, and we have included this information in the introduction of the manuscript (lines 80-82).

 #3. Line 167 - The authors mentioned that “The models and levels of analysis presented in Figure 1 can potentially be expanded to include extra-pharmacological factors…” Is there any potential to incorporate these factors into the current Figure 1?

 We have followed the reviewer's suggestion and integrated the non-pharmacological factors into Figure 1.

#4. Figure 2: Confirm the color code for “Ventromedial prefrontal cortex (vmPFC)” and “Raphe nuclei”.

 Confirmed in figure legend 2 (line 226).

 #5. Minor corrections are needed for the following sentences if applicable:

 Line 123: “The structural and neuroplastic changes brought about by…”

 Corrected (lines 154-155)

 Line 209: “Notably,, ...”

Corrected (line 246)

Round 2

Reviewer 1 Report

Comments and Suggestions for Authors

Bing clarifies the following text, putting it in simple but elegant scientific technical language, without being convoluted: The authors have expanded their review with most of the suggested corrections; when they have not done so, the authors provide convincing arguments for their refusal to include the suggestions. Now the review covers more broadly the recent findings on the use of psychedelic substances as a potential therapeutic agent, as well as their limitations and contraindications.

I only have two observations that I think are necessary:

  1. The authors should elaborate their conclusions based on the information they have provided.
  2. Similarly, they must incorporate the new information into Figure 3. In it, they still do not mention the sigma1 receptor, MAMs, and in a certain way, they still do not explain the activity of psychedelics on GABAergic interneurons, which, as I understand it, would be different from pyramidal neurons. In this way, the function of psychedelics would not be correctly depicted since it does not correspond to the phenomenology observed in aversive motivation, where the activity of psychedelics could be reducing the plasticity associated with BDNF; would psychedelics still activate TRKB in GABAergic interneurons? If so, how do they explain that it reduces plasticity? I think that Figure 3 should be substantially corrected so that the information provided is reflected in it; if they consider it relevant or helpful, the authors can opt to add another figure that explains the reduction of plasticity.

Author Response

We thank the Reviewer for careful review which helped improve the manuscript.

We have addressed and incorporated suggestions. Please find our answer to comments below as well as suggested text changes in the manuscript file with track changes.

Response to Reviewers comments

Bing clarifies the following text, putting it in simple but elegant scientific technical language, without being convoluted: The authors have expanded their review with most of the suggested corrections; when they have not done so, the authors provide convincing arguments for their refusal to include the suggestions. Now the review covers more broadly the recent findings on the use of psychedelic substances as a potential therapeutic agent, as well as their limitations and contraindications.  I only have two observations that I think are necessary:

 The authors should elaborate their conclusions based on the information they have provided.

Similarly, they must incorporate the new information into Figure 3. In it, they still do not mention the sigma1 receptor, MAMs, and in a certain way, they still do not explain the activity of psychedelics on GABAergic interneurons, which, as I understand it, would be different from pyramidal neurons. In this way, the function of psychedelics would not be correctly depicted since it does not correspond to the phenomenology observed in aversive motivation, where the activity of psychedelics could be reducing the plasticity associated with BDNF; would psychedelics still activate TRKB in GABAergic interneurons? If so, how do they explain that it reduces plasticity? I think that Figure 3 should be substantially corrected so that the information provided is reflected in it; if they consider it relevant or helpful, the authors can opt to add another figure that explains the reduction of plasticity.

  We express our gratitude to the reviewer for their positive feedback on our review and the response we submitted. In light of the additional suggestion from the reviewer, we have:

  • Further elaborated on our conclusions, emphasizing the significance of 5-HT2AR, TrkB, Sig-1R pathways and ensuring that the conclusions align more effectively with the information presented throughout the paper (lines 655 – 679).
  • Revised Figure 3 and its corresponding legend, to include the Sig-1R pathway and mitochondria-associated endoplasmic reticulum membranes (MAMs), and ensure that it accurately reflects the information provided in the manuscript.
  • Introduced a new paragraph that offers further clarification regarding the existing knowledge gap concerning the impacts of psychedelics on various types of neurons and their effects on neuroplasticity (lines 405 – 453). Additionally, we have revised the legend for Figure 3 (lines 291-294) to address this knowledge gap. Consequently, we do not find it suitable to include an additional figure based on predominantly speculative and inconclusive data. We have confidence that the information presented in this new paragraph provides a comprehensive perspective on the novelty and scarcity of molecular data concerning the mechanisms involved in neuronal plasticity across different neuron types and brain regions.

 Yours sincerely,

Blerida Banushi

Round 3

Reviewer 1 Report

Comments and Suggestions for Authors

The authors have satisfactorily addressed most of my suggestions. However, I remain puzzled by the observation that psychedelic substances bind to the TrkB receptor, which may account for their antidepressant properties. This contradicts results observed in the mesolimbic system, where blocking the expression of TrkB with a virus inhibits aversion. From my personal observations, animals appear to exhibit superior mood and behavior compared to the control group, unlike when BDNF is injected into the VTA. In the latter case, animals seem to experience a highly distressing state that resembles chronic withdrawal symptoms. I would appreciate it if the authors could discuss this apparent contradiction. Aside from this point, I believe this is an excellent study that makes a significant contribution to the field.

Author Response

We appreciate the reviewer's positive comments and additional suggestions. In order to provide a more comprehensive response to the reviewer's comment, we have elaborated further on the points in question (additional text: lines 371-381 and 403-410).

The challenge of offering a comprehensive response to this question is rooted in the limited availability of molecular data pertaining to the mechanisms the reviewer is inquiring about. While we have proposed potential mechanisms, it's essential to acknowledge that these mechanisms are not yet fully understood, particularly in the context of psychedelics, making them intriguing subjects for future research. It's worth noting that the molecular mechanisms we presented regarding the effects of BDNF-TrkB signaling on enhancing plasticity are well-supported by high-quality studies, as evidenced by the cited literature.

 Furthermore, it is important to emphasize that the reviewer's question, as evident from the additional paragraph we have included, primarily concerns the effects of psychedelics at the neuronal/circuit and brain levels, rather than solely focusing on the known molecular aspects, which is the primary focus of this review. Given the lack of molecular data in the existing literature concerning these effects, we hope that the supplementary information we have provided adequately addresses the reviewer's comment and serves as a reference point for future investigations.